# Analysis of Promoter-Associated Chromatin Interactions Reveals Biologically Relevant Candidate Target Genes at Endometrial Cancer Risk Loci

**DOI:** 10.3390/cancers11101440

**Published:** 2019-09-26

**Authors:** Tracy A. O’Mara, Amanda B. Spurdle, Dylan M. Glubb

**Affiliations:** Department of Genetics and Computational Biology, QIMR Berghofer Medical Research Institute, Brisbane QLD 4006, Australia; tracy.omara@qimrberghofer.edu.au (T.A.O.); amanda.spurdle@qimrberghofer.edu.au (A.B.S.);

**Keywords:** endometrial cancer risk, GWAS, HiChIP, H3K27Ac, chromatin looping, enhancer, promoter

## Abstract

The identification of target genes at genome-wide association study (GWAS) loci is a major obstacle for GWAS follow-up. To identify candidate target genes at the 16 known endometrial cancer GWAS risk loci, we performed HiChIP chromatin looping analysis of endometrial cell lines. To enrich for enhancer–promoter interactions, a mechanism through which GWAS variation may target genes, we captured chromatin loops associated with H3K27Ac histone, characteristic of promoters and enhancers. Analysis of HiChIP loops contacting promoters revealed enrichment for endometrial cancer GWAS heritability and intersection with endometrial cancer risk variation identified 103 HiChIP target genes at 13 risk loci. Expression of four HiChIP target genes (*SNX11*, *SRP14*, *HOXB2* and *BCL11A*) was associated with risk variation, providing further evidence for their targeting. Network analysis functionally prioritized a set of proteins that interact with those encoded by HiChIP target genes, and this set was enriched for pan-cancer and endometrial cancer drivers. Lastly, HiChIP target genes and prioritized interacting proteins were over-represented in pathways related to endometrial cancer development. In summary, we have generated the first global chromatin looping data from normal and tumoral endometrial cells, enabling analysis of all known endometrial cancer risk loci and identifying biologically relevant candidate target genes.

## 1. Introduction

To date, 16 loci have been found to robustly associate with endometrial cancer risk by genome-wide association studies (GWAS) [1,2,3]. As for other GWAS, the vast majority of credible variants (CVs; i.e., the lead variant and other correlated variants) at these loci are non-coding and likely mediate their effects through gene regulation (as reviewed in [4]). Indeed, we previously found that a majority of endometrial cancer risk CVs from ten recently discovered GWAS loci were coincident with promoter- or enhancer-associated epigenetic features in relevant cell lines or tissues [1]. Notably, the overlap between CVs and these elements was significantly greater for features observed in endometrial cancer cell lines after stimulation with estrogen [1], one of the most established risk factors for endometrial cancer [5,6,7].

The follow-up of GWAS is challenging because the target genes of CVs are generally not obvious; particularly as CVs located in enhancers can regulate promoter activity over long distances through chromatin looping [2,8,9,10,11] and enhancers do not necessarily loop to the nearest gene [12]. However, an array of chromatin conformation capture (3C)-based techniques are now available to explore long-range chromatin looping [13] and identify candidate target genes at GWAS loci. These include a local low-throughput 3C approach, which we have previously used to identify chromatin looping between CV-containing regions and *KLF5* at the 13q22.1 endometrial cancer risk locus [2]. Ideally, a high-throughput approach that can simultaneously identify functional elements and interrogate looping between these elements and genes located at endometrial cancer risk loci is required. Hi-ChIP, a technique recently developed from the global Hi-C chromatin looping analysis, appears to fit these requirements. HiChIP can be used to assess chromatin loops associated with specific protein-bound regions, generating high-resolution interactions using fewer cells and fewer sequencing reads than Hi-C [14].

HiChIP has been used to enrich for enhancer–promoter loops by capturing chromatin interactions associated with H3K27Ac [12], a histone mark characteristic of active promoters and enhancers. The value of H3K27Ac in identifying likely functional GWAS variation is further evidenced by many reports of enrichment of GWAS-identified variation in H3K27Ac-associated genomic regions [10,15,16,17]. Moreover, the ability of H3K27Ac HiChIP to facilitate the identification of candidate target genes at GWAS loci has been demonstrated in two recent studies [12,18]. 

Global chromatin looping analysis of endometrial cells has not been previously reported. In the current study, we have performed H3K27Ac HiChIP in normal and tumoral endometrial cells to capture enhancers or promoters that overlap with endometrial cancer GWAS risk CVs. We have then used the H3K27Ac HiChIP data to identify genes likely to be targeted by these regulatory elements through chromatin interactions and analysed these candidate target genes to aid the biological interpretation of endometrial cancer GWAS risk variation.

## 2. Results

### 2.1. H3K27Ac HiChIP Analysis Identifies Promoter-Associated Chromatin Loops in Endometrial Cell Lines

To assess chromatin looping at endometrial cancer risk loci, we sequenced and analysed H3K27Ac HiChIP libraries from a normal immortalized endometrial cell line (E6E7hTERT) and three endometrial cancer cell lines (ARK1, Ishikawa and JHUEM-14) for valid chromatin interactions (Appendix A). We identified 66,092 to 449,157 cis HiChIP loops (5 kb–2 Mb in length) per cell line, with a majority involving interactions of over 20 kb in distance (Table 1). Of the total loops, 35%–40% had contact with a promoter and these promoter-associated loops had a median span >200 kb (Table 1), indicating that they may be involved with long-range gene regulation. BED files for promoter-associated loops can be found in Appendix A.

### 2.2. HiChIP Promoter Loops Are Enriched for Endometrial Cancer Heritability

To determine if promoter-associated HiChIP loops (i.e., those potentially involved with gene regulation) are enriched for heritability of endometrial cancer at a genome-wide level, we applied stratified linkage disequilibrium (LD) score regression analysis to the GWAS summary statistics from the largest study of endometrial cancer performed to date [1]. All four endometrial cell lines demonstrated an enrichment of endometrial cancer heritability in the anchors of promoter-associated loops (Table 2), although the enrichment in Ishikawa loops did not reach statistical significance (Bonferroni threshold, *p* < 0.0125).

### 2.3. HiChIP Promoter Looping Reveals 103 Candidate Target Genes at Endometrial Cancer Risk Loci

To identify HiChIP target genes at the 16 known endometrial cancer GWAS loci, we intersected CVs with HiChIP promoter-associated loops from the four endometrial cell lines. Through this analysis, we identified 103 HiChIP target genes (81 protein coding and 22 non-coding; Table 3 and Appendix A) at 13 endometrial cancer GWAS risk loci. Ten of the non-coding HiChIP target genes encoded anti-sense transcripts (Appendix A), with these genes often sharing a promoter with a coding HiChIP target gene, e.g., *CDKN2A* and *CDKN2B-AS1* (9p21.3; Figure 1) and *WT1* and *WT1-AS* (11p13; Appendix A).

The number of HiChIP target genes at a locus ranged from 1 (2p16.1 and 12q24.21) to 38 (17q21.32; Appendix A), with a median of 4 (Table 3; also see the representative examples of chromatin looping at loci in Figure 1 and Appendix A). The HiChIP target genes included three genes (*WT1*, *WT1-AS* and *GNL2*) that had CVs located in a HiChIP looping contact at their promoter region, but for which there was no looping from an element containing a distal CV (e.g., *WT1* and *WT1-AS* in Appendix A). These findings suggest that *WT1*, *WT1-AS* and *GNL2* may be directly regulated by a promoter CV in endometrial cells. In total, only 18 genes were the nearest gene to a CV (Table 1). More than one-third (36%) of the HiChIP target genes were identified using looping data from at least two endometrial cell lines (underlined in Table 3; Appendix A), providing additional evidence for their targeting.

### 2.4. HiChIP Target Genes Are Enriched for Potential Targets of a Mirna Encoded by the HiChIP Target Gene Mir196a1

ToppFun bioinformatic analysis revealed that the predicted targets of 86 miRNAs were over-represented among the HiChIP target genes (*p*_Bonferroni_ < 0.05, Appendix A). hsa-mir-196a-5p was one of these miRNAs and is encoded by *MIR196A1*, itself one of the seven miRNA genes among the HiChIP-identified targets (Table 3). hsa-mir-196a-5p is predicted to bind to transcripts from six HiChIP target genes: four of which (*HOXB1*, *HOXB6*, *HOXB7* and *HOXB8*) are located at the same locus as *MIR196A1* (17q21.32), with the remaining two (*BRAP* and *RASGRP1*) encoded at endometrial cancer risk loci on other chromosomes (12q24.11 and 15q15.1, respectively).

### 2.5. HiChIP Target Genes Are Differentially Expressed in Endometrial Tumors

HiChIP target genes were evaluated for significant differential gene expression in endometrial tumor and paired normal samples (The Cancer Genome Atlas Project (TCGA) [19]). Thirty-six HiChIP target genes were found to be differentially expressed in tumors, with 17 genes down-regulated and 19 up-regulated (Appendix A). *TNFAIP8L3*, encoding a lipid transfer protein [20], was the most strongly down-regulated gene in tumors (log_2_ fold-change = −5.1, q = 1.24 × 10^−133^; Appendix A) and *SPINT1*, encoding a serine peptidase inhibitor [21], the most strongly up-regulated gene (log_2_ fold-change = 5.8, q = 1.52 × 10^−93^; Appendix A). Statistical assessment of HiChIP target genes by Fisher’s exact test revealed a more than two-fold over-representation of genes that were differentialy expressed in endometrial tumors (OR = 2.39, 95% CI 1.59–3.59, p = 6.08 × 10^−05^).

### 2.6. HiChIP Target Gene Expression Associates with CVs

To aid prioritisation of HiChIP target genes, we interrogated expression quantitative trait locus (eQTL) data from the largest study of whole-blood gene expression [22] and TCGA endometrial tumors [23]. Using these data, we evaluated the overlap between endometrial cancer risk CVs and the top eQTL variants for each HiChIP target gene. From the blood eQTL data, we found that the lead CV at the 17q21.32 risk locus, rs882380, was one of the top eQTLs for *SNX11* and *HOXB2*, and the lead CV at the 15q15.1 risk locus, rs937213, was one of the top eQTLs for *SRP14* (Table 4 and Appendix A). From the endometrial tumor eQTL data, we found that the top eQTL for the HiChIP target gene *BCL11A* was rs7579014, a CV at the 2p16.1 risk locus (Table 4 and Appendix A).

### 2.7. Protein-Protein Interaction Network of HiChIP Target Genes Reveals Enrichment for Endometrial Cancer Driver Genes

The HiChIP target genes included three known pan-cancer driver genes (*CDKN2A*, *TBX3* and *WT1*) identified by Bailey et al. [24], but no known drivers of endometrial cancer from lists compiled by Bailey et al. or Gibson et al. [25]. Pathway analysis was performed by ToppFun using the 103 candidate target genes to gain biological insights but no pathways were found to be enriched after Bonferroni correction. To explore protein–protein interaction networks involving the candidate target genes, we used the ToppGenet bioinformatic tool. Mining of protein–protein interaction databases by ToppGenet revealed 2135 proteins that interacted with those encoded by HiChIP target genes (Appendix A). Prioritisation was then performed by ToppGenet to identify those proteins with the most similar functional features to the HiChIP target gene set, i.e., a “guilt by association” approach. Using this method, 387 of the interacting proteins had significant similarity scores at a 5% false discovery rate (FDR) (Appendix A). The protein with the most statistically significant similarity score was encoded by *TP53*, an endometrial and pan-cancer driver gene. Indeed, many other proteins encoded by known cancer driver genes were observed in the prioritised set of proteins. Of the 85 pan-cancer driver genes encoding interacting proteins, 55 were observed in the prioritised set, a significant enrichment (OR = 9.49, 95% CI 6.06–14.80; *p* < 1 × 10^−09^; Appendix A). The two available lists of endometrial cancer driver genes [24,25] were combined and of the 28 encoding interacting proteins, 19 were observed in the prioritised set (Table 4; Appendix A), also a significant enrichment (OR = 9.98, 95% CI 4.64–22.58; *p* = 1.8 × 10^−08^).

### 2.8. HiChIP Target Genes and Interacting Proteins Are Over-Represented in Relevant Biological Pathways

Pathway analysis using the combined list of 103 HiChIP target genes and 387 prioritised interacting proteins found 462 pathways to be significantly enriched after Bonferroni correction (Appendix A). Many of these pathways were related to gene regulation (e.g. “transcriptional misregulation in cancer”) and cancer (e.g., “pathways in cancer”), including hallmarks of cancer identified by Hanahan and Weinberg [26] (Table 5). A KEGG “endometrial cancer” pathway and pathways related to endometrial cancer risk factors, such as obesity (e.g., “signaling by leptin”), insulinemia (e.g., “insulin receptor signalling cascade”) and estrogen exposure (e.g.,“plasma membrane estrogen receptor signalling”) were also found among the significantly enriched pathways.

## 3. Discussion

We performed H3K27Ac HiChIP in endometrial cell lines to enrich for enhancer–promoter chromatin looping interactions and found more than a third of identified HiChIP chromatin loops interacted with a promoter. The anchors of promoter-associated loops from two endometrial cancer cell lines and an immortalised normal endometrial cell line were significantly enriched for endometrial cancer heritability, highlighting the potential importance of these loops in mediating the effects of endometrial cancer risk variation. Intersection of the promoter-associated loops with endometrial cancer GWAS CVs revealed 103 HiChIP target genes at 13 loci, 36% of which were identified from loops in multiple cell lines. At only two loci (2p16.1 and 12q24.21) was the nearest gene the only HiChIP target identified (*BCL11A* and *TBX3*, respectively). Similar to another HiChIP study that integrated chromatin interaction data with GWAS findings, the majority of HiChIP target genes (83%) involved a CV-promoter looping interaction that skipped the gene(s) closest to the CVs at that locus [12]. These findings further highlight the potential for long-range regulation and the pitfalls of mapping GWAS variation to the nearest gene (as we have previously discussed [27]). The median lengths of the promoter-associated loops in the four HiChIP cell lines were all greater than 200 kb, consistent with gene skipping by the putative enhancers at the endometrial cancer risk loci. Furthermore, the observed rate of gene skipping was similar that found by HiChIP analyses of other disease risk loci [12].

The HiChIP target genes were enriched for genes that are differentially expressed in endometrial tumors, providing evidence that these genes may be involved with endometrial cancer development and that the HiChIP data can help identify biologically relevant genes. Moreover, of the 25 candidate target genes previously identififed from endometrial cancer GWAS (reviewed in [4]), the targeting of ten genes (*MIR1207*, *WT1-AS*, *RCN1*, *SH2B3*, *BMF*, *GPR176*, *SRP14-AS1*, *SRP14*, *HNF1B* and *SNX11*) was supported by the HiChIP data. Anoher of the previously identified candidate target genes was *KLF5* at the 13q22.1 risk locus. Endometrial cancer GWAS risk variation has been found to loop to *KLF5* in data generated by a local 3C-based technique [2], an interaction which we did not observe in our HiChIP analyses. Our HiChIP approach had much higher resolution (due to the use of a restriction enzyme producing smaller fragments) and closer examination of the HiChIP data at 13q22.1 revealed a *KLF5* promoter interaction in JHUEM-14 cells that looped to an anchor 23 bp from endometrial cancer risk CV rs9600103. 

Assessment of the other two loci without HiChIP target genes, 6q22.3 and 6q22.31, did not reveal other promoter loops in such close proximity to CVs. It is possible, at these and other loci, that additional genes may be targeted through chromatin looping occurring in cell types (or settings) not studied here. Indeed, a limitation of this study is the use of cell lines to model chromatin looping that occurs in normal or tumoral endometrial tissue. For example, chromatin looping patterns could have been altered by immortalization (i.e., of the normal endometrial epithelial cells used to generate the E6E7hTERT cell line), culture conditions (e.g., two-dimesional culture on plates) or incorporation of mutations during passaging. Another limitation is that only chromatin looping associated with H3K27Ac was captured by our approach, whereas the effects of some endometrial cancer risk variation may relate to other epigenomic features or mechanisms (e.g., methylation). HiChIP analysis of tissue or primary cells, or alternative functional genomic approaches (e.g., CRISPR genome/epigenome editing) may thus prove useful in identifying further candidate target genes (particularly at 6q22.3, 6q22.31 and 13q22.1). Furthermore, these approaches would aid prioritisation and validation of candidate target genes identified in this study.

To provide evidence to support the regulation of HiChIP target genes by endometrial cancer risk CVs, we integrated available eQTL data and found that CVs associated with the expression of four HiChIP target genes. Three of these associations were observed in whole blood (*SNX11*, *HOXB2* and *SRP14*) and one in endometrial tumors (*BCL11A*). *SNX11* encodes a member of the sorting nexin family and is involved in endosomal intracellular trafficking [28] and may prevent degradation of its endosomal cargo [29]. *HOXB2* is a homeobox B gene and encodes a transcription factor that is involved in development in mice [30]. Expression of *HOXB2* (along with the HiChIP target genes *HOXB5* and *PTHLH*) has been found to be downregulated in a rare syndrome that is characterised by abnormal development of the uterus and vagina [31]. In our study, we found that reduced *HOXB2* expression in whole blood was associated with endometrial cancer risk variation, compatible with reports that HOXB2 has a tumor-suppressor function [32,33]. SRP14 is involved in the formation of stress granules [34], which can also be initiated by phosphorylation of the protein encoded by the *EIF2AK4* HiChIP target gene [35]. Consistent with our observation that endometrial cancer risk CVs were associated with increased *SRP14* expression, stress granules promote cell survival and cancer cell fitness, and their components are upregulated in tumors (reviewed in [36]). Finally, *BCL11A* encodes a zinc finger protein transcription factor and plays an important role in lymphocyte development [37]. In cancer, the role of BCL11A protein appears to be context-dependent. In some cancers, it has oncogenic effects [38,39], whereas in T cell leukaemia, it may act as a tumor suppressor [37]. Further supporting a tumor-suppressor role are findings that down-regulation of *BCL11A* increases the resistance of cancer cells to radiation [40] and loss of function of *BCL11A* is associated with genome instability in lung cancer [41]. Concordant with an anti-cancer function, we found that endometrial cancer risk CVs were associated with decreased *BCL11A* expression in endometrial tumors.

Six HiChIP target genes (*BRAP*, *RASGRP1*, *HOXB1*, *HOXB6*, *HOXB7* and *HOXB8*) were enriched for miRNA targets of *MIR196A1*, itself a HiChIP target gene, providing evidence to link these genes together in a potential network that may mediate the effects of endometrial cancer risk variation. *MIR196A1* miRNA has also been shown to regulate *HOXB9*, another HiChIP target gene [42]. Relevantly, expression of *MIR196A1* miRNA has been correlated with expression of the endometrial cancer candidate target gene *KLF5* in breast tumors, and has been associated with poor outcome in breast and ovarian cancer patients [43,44]. *MIR196A1* miRNA also inhibits a range of cancer cell phenotypes including apoptosis, proliferation, migration and invasion [42,45,46,47]. Consistent with these observations, *MIR196A1* miRNA levels are lower in endometrial tumors compared with healthy endometrial tissue [47]. There may also be a link between *MIR196A1* and the endometrial cancer risk factors of obesity and insulinemia [48,49]: *MIR196A1* miRNA is upregulated in gluteofemoral fat, which is associated with lower risk of diabetes [50]; and forced expression of the mouse homologue of *MIR196A1* has been found to make mice resistant to obesity and prevent them from developing insulin resistance [51].

Lastly, bioinformatic analysis was used to functionally prioritize 387 proteins that interact with those encoded by the HiChIP target genes. This prioritized set had nearly a ten-fold over-representation of proteins encoded by pan-cancer or endometrial cancer driver genes compared to other interacting proteins. Further analysis demonstrated that the combined set of prioritised proteins and candidate target genes was enriched for pathways relevant to hallmarks of cancer and endometrial cancer risk factors. Taken together, these observations suggest that proteins encoded by HiChIP target genes may mediate their effects through interactions with cancer drivers and other proteins that are involved in endometrial cancer development pathways.

## 4. Materials and Methods

### 4.1. Cell Culture

Ishikawa, JHUEM-14, ARK-1 and E6E7hTERT cells were a gift from Prof PM Pollock (Queensland University of Technology). Cell lines were authenticated using STR profiling and confirmed to be negative for mycoplasma contamination. For routine culture, Ishikawa, ARK-1 and E6E7hTERT cells were grown in Dulbecco’s modified Eagle’s medium (DMEM) with 10% fetal bovine serum (FBS) and antibiotics (100 IU/mL penicillin and 100 μg/mL streptomycin). JHUEM-14 cells were cultured in DMEM/F12 medium with 10% FBS and antibiotics. All cell lines were cultured in a humidified incubator (37 °C, 5% CO_2_).

### 4.2. Cell Fixation

For fixation, cells on 10 cm tissue culture plates (~80% confluence) were washed with PBS and fixed at room temperature in 1% formaldehyde in PBS. After 10 min, cells were placed on ice and the formaldehyde was quenched by washing twice with 125 mm glycine in PBS. Cells were removed from the dish with a cell scraper and washed with PBS before the storage of cell pellets at −80 °C. As we had previously observed greater enrichment of endometrial cancer GWAS variation in epigenomic features observed after estrogen stimulation, including those found in Ishikawa and JHUEM-14 endometrioid endometrial cancer cell lines [1], these two cell lines were stimulated with 10 nm estradiol for 3 h prior to fixation, as per [1]. Normal endometrium is considered estrogen-responsive [52], so E6E7hTERT (a normal immortalised endometrial cell line) cells were also stimulated with estradiol; whereas, ARK-1 cells (derived from a serous endometrial tumor) were not stimulated with estradiol as serous endometrial tumors are not considered to be estrogen-responsive [53]. 

### 4.3. HiChIP Library Generation

HiChIP libraries were generated as per the method of Mumbach et al. [14] with modifications. Briefly, cell nuclei were extracted from fixed cell pellets and digested overnight with 375U of DpnII to improve resolution [54]. After digestion, nuclei were resuspended in NEB buffer 2 (New England Biolabs) and restriction fragment overhangs were filled-in with biotin-dATP using the DNA polymerase I, large Klenow fragment (incubated at 30 °C for 2 h). Proximity ligation was performed for 4 h at 16 °C, then nuclei lysed and chromatin sheared for 9 min using the Covaris S220 Sonicator as per Mumbach et al. For each sample, sheared chromatin was split into two tubes and incubated overnight with 4.6 μg of H3K27Ac antibody (Abcam, EP16602). The next day, Protein A beads were used to capture H3K27Ac-associated chromatin, which was eluted and purified with Zymo Research concentrator columns (columns were washed twice with 10 μL of water). As per Mumbach et al., the DNA concentration of the purified chromatin was to estimate the amount of TDE1 enzyme (Illumina) needed for tagmentation, which was performed with biotin-labelled chromatin captured on streptavidin beads. Sequencing libraries were then generated by PCR of tagmented samples using the Nextera DNA preparation kit (Illumina) as per the manufacturer’s instructions. Afterwards, size selection was performed using Ampure XP beads to capture 300–700 bp fragments. For each cell line, at least two independent sequencing libraries were pooled together to provide 25 μL of library at ≥10 nm for Illumina HiSeq4000 (AGRF, Brisbane, QLD, Australia) paired-end sequencing.

### 4.4. HiChIP Bioinformatic Analyses

HiChIP reads (fastq files) were aligned to the human reference genome (hg19) using HiC-Pro version 2.9.0 [55] and default settings used to remove duplicate reads, assign reads to DpnII restriction fragments and filter for valid interactions. The hichipper pipeline version 0.7.0 [56] was used to process all valid reads from HiC-Pro, with the HiChIP reads used to identify H3K27Ac peaks using the standard MACS2 background model. Chromatin interactions were filtered using a minimum distance of 5 kb and a maximum of 2 Mb. The final set of chromatin loops used for further investigation were interactions which were supported by a minimum of two unique paired-end tags and with a Mango [57] *q*-value < 5%.

### 4.5. Stratified LD Score Regression Analysis

We used stratified LD score regression [58,59] to quantify enrichment of endometrial cancer risk variation in HiChIP promoter-associated loops. Stratified LD score regression calculates enrichment as the proportion of genetic heritability attributable to a particular set of variants (e.g., variants located within HiChIP promoter-associated loops) divided by the proportion of total genetic variants annotated to that set. Enrichment for each cell line HiChIP promoter-associated loops annotation categories were assessed individually conditional on a “full baseline model” of 53 overlapping categories as used previously (https://data.broadinstitute.org/alkesgroup/LDSCORE/) [59]. For regression, variants were pruned to the HapMap3 variant list (~1 million variants) and the 1000 Genomes Project Phase 3 European population variants were used for the LD reference panel. The major histocompatibility complex (MHC) region was removed from this analysis because of its complex LD structure.

### 4.6. Identification and Analysis of HiChIP Target Genes

Promoter-associated loops were defined as HiChIP loops with an anchor within 3 kb of a transcription start site (GRCh37; accessed May 2019). To identify candidate target genes, HiChIP promoter loops were intersected with endometrial cancer risk CVs (*n* = 457) which had been determined using a 100:1 log likelihood ratio with the p-value for the lead variant at each GWAS locus. Differential gene expression from TCGA endometrial tumor (*n* = 174) and normal (*n* = 13) samples was obtained from GEPIA2 [60] using limma analysed data with a log_2_ fold cut-off of 1 and *q* < 0.01 for statistical significance. All bioinformatic analysis of HiChIP target genes was performed using the ToppGene Suite of tools (accessed 7 June 2019) [61]. ToppFun was used to detect enrichment of gene lists based on miRNA binding sites and pathways using hypergeometric distribution analysis. ToppGenet was used to identify and prioritise genes in protein–protein interaction networks based on functional similarity to the HiChIP target genes. Analyses to identify over-representation of genes in different sets was performed using Fisher’s exact test in GraphPad Prism 8.1.2.

## 5. Conclusions

Here, we present the first global study of chromatin looping in endometrial cell lines, using an H3K27Ac HiChIP approach to enrich for enhancer–promoter interactions. These data will provide an extremely useful resource for genetic studies of not only endometrial cancer but also other diseases that involve the endometrium. Through these data, we have found that promoter-associated HiChIP loops are significantly enriched for endometrial cancer heritability and used these loops to identify a set of candidate target genes at endometrial cancer GWAS loci, which contains an over-representation of genes differentially regulated in endometrial tumors. Integration of eQTL data provided evidence to prioritize candidates for functional studies and further supports the hypothesis that endometrial cancer GWAS variation regulates gene expression through long-range regulatory interactions. Previous reports from the literature suggests there is interplay among the products of HiChIP target genes and that proteins encoded by HiChIP target genes interact with cancer drivers. Finally, bioinformatic analysis indicates that the HiChIP target genes and their interacting protein–protein networks belong to pathways that are relevant to endometrial cancer development.

In summary, this study has identified candidate endometrial cancer GWAS target genes for future studies and furthers our understanding of the genetic basis of endometrial cancer development.

## Figures and Tables

**Figure 1 cancers-11-01440-f001:**
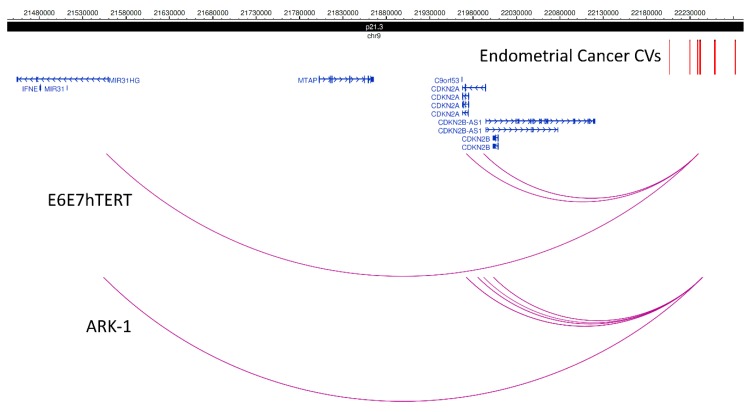
Promoter-associated chromatin looping identifies HiChIP target genes at the 9p21.3 locus. Promoter-associated loops were intersected with endometrial cancer risk CVs (coloured red), revealing loops (purple) that interact with the promoters of *MIR31HG* (in E6E7hTERT and ARK-1 cells), *CDKN2A* (in E6E7hTERT and ARK-1 cells), *CDNK2B-AS1* (in E6E7hTERT and ARK-1 cells) and *CDKN2B* (in ARK-1 cells). *CDK2NA* and the non-coding anti-sense gene CDKN2B-AS1 are encoded on opposite DNA strands and share a promoter.

**Table 1 cancers-11-01440-t001:** Characteristics of HiChIP loops in the endometrial cell lines.

Cell Line	Total Loops	Loops < 20 kb	Loops > 20 kb	Promoter-Asociated Loops	Median Span of Promoter-Associated Loops (kb)
E6E7hTERT	162,476	25,133 (15.5%)	137,343 (84.5%)	59,658 (36.7%)	206
ARK1	449,157	45,932 (10.2%)	403,225 (89.8%)	155,080 (34.5%)	282
Ishikawa	219,067	29,954 (13.7%)	189,113 (86.3%)	79,309 (36.2%)	259
JHUEM-14	66,092	10,254 (15.5%)	55,838 (84.5%)	26,492 (40.0%)	209

**Table 2 cancers-11-01440-t002:** HiChIP promoter-associated loops in endometrial cell lines are enriched for endometrial cancer heritability.

Cell Line	Enrichment (Standard Error)	*p*-Value
E6E7hTERT	6.92 (1.70)	1.30 × 10^−04^
ARK1	4.08 (0.84)	2.50 × 10^−04^
JHUEM-14	9.61 (3.11)	5.00 × 10^−03^
Ishikawa	3.23 (1.18)	0.07

**Table 3 cancers-11-01440-t003:** HiChIP target genes at endometrial cancer risk loci.

Risk Locus	HiChIP Target Genes	Nearest Gene(s) to CVs ^1^
1p34.3	*GNL2*, *C1orf122*	*GNL2*, *RSPO1*
2p16.1	*BCL11A*	*BCL11A*
8q24.1	*MIR1207*, *PVT1*, *LINC00824*	*LINC00824*
9p21.3	*CDKN2A*, *CDKN2B*, *CDKN2B-AS1*, *MIR31HG*	*CDKN2B-AS1*
11p13	*WT1*, *WT1-AS*, *CD59*, *PAX6*, *RCN1*	*WT1-AS*
12p12.1	*BHLHE41*, *PTHLH*, *SSPN*, *LRMP*	*SSPN*
12q24.11	*SH2B3*, *PHETA1*, *ACAD10*, *ARPC3*, *BRAP*, *IFT81*, *LINC02356*	*SH2B3*, *ATXN2*
12q24.21	*TBX3*	*TBX3*
15q15.1	*SRP14*, *SRP14-AS1*, *BMF*, *BAHD1*, *CCDC9B*, *GPR176*, *KNSTRN*, *PAK6*, *PLCB2*, *PLCB2-AS1*, *THBS1*, *EIF2AK4*, *CHST14*, *DISP2*, *FSIP1*, *INAFM2*, *PLA2G4B*, *RASGRP1*, *SPINT1*, *ANKRD63*, *PHGR1*, *SPINT1-AS1*, *C15orf56*	*SRP14*, *SRP14-AS1*, *EIF2AK4*
15q21.2	*DMXL2*, *TRPM7*, *TNFAIP8L3*	*CYP19A1*
17q11.2	*RAB11FIP4*, *MIR193A*, *TEFM*, *RNU6ATAC7P*	*RAB11FIP4*, *NF1*, *EVI2A*, *EVI2B*
17q12	*HNF1B*, *DUSP14*, *MRM1*, *MRPL45*, *SRCIN1*, *TBC1D3*, *C17orf78*	*HNF1B*
17q21.32	*SNX11*, *MIR1203*, *SKAP1-AS1*, *SKAP1*, *CBX1*, *HOXB1*, *HOXB2*, *HOXB3*, *HOXB4*, *HOXB5*, *HOXB6*, *HOXB7*, *HOXB8*, *HOXB9*, *HOXB13*, *HOXB-AS1*, *HOXB-AS3*, *HOXB-AS4*, *PRR15L*, *CDK5RAP3*, *LRRC46*, *MRPL10*, *NFE2L1*, *SCRN2*, *CALCOCO2*, *COPZ2*, *DLX3*, *KPNB1*, *PNPO*, *SNF8*, *SP2*, *SP2-AS1*, *SP6*, *MIR10A*, *MIR152*, *MIR196A1*, *MIR3185*, *PHOSPHO1*	*SNX11*, *MIR1203*, *SKAP1-AS1*, *SKAP1*, *CBX1*

^1^ At some loci, CVs are coincident with multiple genes. Underlined candidate target genes are supported by HiChIP data from multiple cell lines.

**Table 4 cancers-11-01440-t004:** Endometrial cancer drivers interacting with proteins encoded by HiChIP target genes.

Protein Encoding Gene	Similarity Score	*p*-Value	FDR ^1^ Value
*TP53*	0.60	3.65E−09	4.00E−06
*ESR1*	0.54	5.95E−07	1.27E−04
*FOXA2*	0.57	1.86E−06	1.99E−04
*EP300*	0.41	8.40E−06	4.27E−04
*CTNNB1*	0.47	1.35E−05	5.54E−04
*PTEN*	0.46	1.98E−05	7.05E−04
*CCND1*	0.49	2.12E−05	7.42E−04
*FGFR2*	0.44	3.97E−05	1.10E−03
*RB1*	0.50	8.21E−05	1.91E−03
*MYCN*	0.44	1.15E−04	2.51E−03
*ERBB2*	0.39	4.15E−04	6.28E−03
*AKT1*	0.35	5.24E−04	7.31E−03
*ERBB3*	0.39	1.12E−03	0.01
*MAX*	0.31	1.75E−03	0.02
*NRIP1*	0.32	1.82E−03	0.02
*ATM*	0.31	2.05E−03	0.02
*CHD4*	0.34	2.67E−03	0.02
*FBXW7*	0.38	3.75E−03	0.03
*DICER1*	0.33	4.44E−03	0.03
*KRAS*	0.33	9.91E−03	0.05
*TAF1*	0.27	0.03	0.11
*ATR*	0.29	0.04	0.13
*PIK3R2*	0.19	0.06	0.17
*POLE*	0.26	0.07	0.17
*CHD3*	0.20	0.14	0.26
*TAB3*	0.22	0.36	0.45
*METTL14*	0.21	0.40	0.49
*KANSL1*	0.09	0.67	0.67

^1^ False-discovery rate (FDR). Bolded proteins have a statistically significant similarity score (FDR < 0.05)

**Table 5 cancers-11-01440-t005:** Examples of enriched pathways related to hallmarks of cancer.

Cancer Hallmark	Related Pathway (Source)	p_Bonferroni_
*Evading growth suppressors*	Regulation of TP53 activity (REACTOME)	1.44E−07
*Avoiding immune destruction*	Innate immune system (REACTOME)	2.06E−06
*Enabling replicative immortality*	Regulation of telomerase (Pathway Interaction Database)	1.43E−12
*Tumor-promoting inflammation*	Inflammation mediated by chemokine and cytokine signalling pathway (PantherDB)	0.03
*Activating invasion and metastasis*	Focal adhesion (KEGG)	5.61E−15
*Inducing angiogenesis*	VEGFA-VEGFR2 pathway (REACTOME)	9.10E−08
*Genome instability and mutation*	RB Tumor Suppressor/Checkpoint Signaling in response to DNA damage (MSigDB C2 BIOCARTA)	1.05E−04
*Resisting cell death*	Apoptosis signaling pathway (Panther DB)	1.78E−08
*Deregulating cellular energetics*	Choline metabolism in cancer (KEGG)	2.10E−04
*Sustaining proliferative signalling*	PI3K-Akt signalling pathway (KEGG)	1.15E−18

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
