# Peer review of "Analysis of Promoter-Associated Chromatin Interactions Reveals Biologically Relevant Candidate Target Genes at Endometrial Cancer Risk Loci"

_cancers, 2019, doi:10.3390/cancers11101440_

Round 1
Reviewer 1 Report
Comments for Cancers-594954
In this paper, authors have used H3K27Ac HiChIP to identify likely regulatory promoter-associated chromatin interactions in normal and tumoral endometrial cells. Based on these data, author assessed promoter associated chromatin loops for enrichment of endometrial cancer heritability and identified candidate target genes of CVs at endometrial cancer risk loci. Author suggest that proteins encoded by HiChIP target genes may mediate their effects through interactions with cancer drivers and other proteins that are involved in endometrial cancer development pathways. Although the study topic is interesting, this study just showed a profile of target gene by analysis of HiChip in endometrial cancer risk loci. Besides the genetic profile presented by the author have not been accurately verified as a risk factor in endometrial cancer. Thus, in my opinions, the ensemble of the presented data do not fully support the conclusions.
Minor Comments
*Although H3K27Ac have known as a histone mark characteristic of active promoter and enhancers and were elevated in various cancer types, it does not consider as risk factor for endometrial tumor development. Author should clearly explain in the INTRODUCTION section why select H3K27AC for identify a potential risk gene at endometrial cancer risk loci.
Major Comments
*Author showed that HICHIP promoter looping reveals 103 candidate target genes at endometrial cancer risk loci (Line 83, table3) and described about HICIP target gene. HiChIP target genes are enriched for potential targets of a miRNA encoded by the HiChIP target gene (Lines108, table3). I wonder why they suddenly mentioned target genes of miRNA. Do the authors mean that potential targets of a miRNA encoded by the HiChIP target gene is risk factor in endometrial cancer development?
* Author analyzed H3K27AC HiChIP libraries from a normal immortalized endometrial cell lines and three endometrial cancer cell lines. They confirmed these data pattern in endometrial tumor and paired normal samples from TCGA data (Supplementary Table S2). Although HiChIP target genes were differentially expressed in endometrial tumor. I don't think these results match well with cell-based results (eg., MIR196A1, HOXB6, HOXB7..).
Author Response
Minor Comments
*Although H3K27Ac have known as a histone mark characteristic of active promoter and enhancers and were elevated in various cancer types, it does not consider as risk factor for endometrial tumor development. Author should clearly explain in the INTRODUCTION section why select H3K27AC for identify a potential risk gene at endometrial cancer risk loci.
We apologize for any ambiguity with our approach but, to clarify, we do not consider H3K27Ac as a risk factor for endometrial tumor development. However, as many studies have demonstrated enrichment of GWAS variants in H3K27Ac-associated regions, we hypothesized that: i) endometrial cancer GWAS risk variation in these regions may be important in mediating the effects of risk variants; and ii) chromatin looping from H3K27Ac-associated regions would point to relevant target genes at endometrial cancer risk loci. We have added text to the Introduction (lines 54-58) explaining our rationale.
Major Comments
*Author showed that HICHIP promoter looping reveals 103 candidate target genes at endometrial cancer risk loci (Line 83, table3) and described about HICIP target gene. HiChIP target genes are enriched for potential targets of a miRNA encoded by the HiChIP target gene (Lines108, table3). I wonder why they suddenly mentioned target genes of miRNA. Do the authors mean that potential targets of a miRNA encoded by the HiChIP target gene is risk factor in endometrial cancer development?
We thought it was notable that the HiChIP candidate genes were enriched for potential targets of a miRNA, which is itself encoded by a HiChIP candidate gene. However, we do not think that these genes are risk factors per se (e.g. akin to BRCA1 in breast cancer). Instead, this finding provides evidence to link these genes together in a potential network which may mediate the effects of endometrial cancer risk variation (text to this effect has been added to the Discussion – lines 247-250)
* Author analyzed H3K27AC HiChIP libraries from a normal immortalized endometrial cell lines and three endometrial cancer cell lines. They confirmed these data pattern in endometrial tumor and paired normal samples from TCGA data (Supplementary Table S2). Although HiChIP target genes were differentially expressed in endometrial tumor. I don't think these results match well with cell-based results (eg., MIR196A1, HOXB6, HOXB7..).
To clarify, we do not believe that the differential expression between tumors and paired normal samples confirms the HiChIP data. Rather, we found a significant enrichment of the differentially expressed genes among the HiChIP target genes, demonstrating the ability of the HiChIP approach to identify biologically relevant genes, as discussed (lines 198-200). We are not sure what the reviewer means when the say these results don’t match with the ‘cell-based results’ in reference to MIR196A1 and its potential target genes (e.g. HOXB6 and HOXB7). Although we have shown that HOXB6 and HOXB7 are up-regulated in endometrial tumors (Supplementary Table S2), it is not possible from the HiChIP data to predict the directionality of any potential effect of endometrial cancer risk variation on gene expression.
Reviewer 2 Report
The authors performed HiCHIP on 4 different endometrial cell lines, one normal, 3 cancer cell line, and integrate the result with GWAS and eQTL to identify target genes. The analyses were properly conducted and results are well presented.
Some minor comments:
Are there any difference of HiCHIP promoter loops between the normal cell line and the 3 cancer cellines? Any specific signal for cancer cell line? If yes, are these target genes more enriched in cancer ? Maybe a good idea to discuss about the limitation of this study. For example, there may be difference between celline and real tissue.Author Response
Are there any difference of HiCHIP promoter loops between the normal cell line and the 3 cancer cell lines? Any specific signal for cancer cell line? If yes, are these target genes more enriched in cancer?
Of the 103 target genes, four were identified solely from loops in the normal cell line and 77 solely from loops in at least one cancer cell line. Analyzing the protein-protein interaction network for the target genes specific to looping in cancer cells, we find there is less enrichment of pan-cancer driver genes (OR=4.40, 95%CI 2.83-6.81) compared to the network for the total set of target genes (OR=9.49, 95%CI 6.06-14.80). For the endometrial cancer driver genes, there is greater enrichment in the network generated from the cancer-specific target genes (OR=14.25, 95%CI 6.10-37.23) compared to the network from generated from the total set of target genes (OR=9.98, 95%CI 4.64-22.58); however, there is substantial overlap between the 95%CIs for these enrichment estimates. Therefore, focusing on target genes identified only in cancer cell lines does not appear to significantly enrich for endometrial cancer drivers and, if anything, reduces the enrichment of pan-cancer .
Maybe a good idea to discuss about the limitation of this study. For example, there may be difference between cell line and real tissue.
We have added further discussion of the limitations of this study in lines 213-220.
Reviewer 3 Report
Using H3K27ac HiChIP in cell line models of endometrial cancer and normal endometrial cells, O’Mara and colleagues aimed to establish putative functional gene targets for GWAS endometrial cancer risk variants. The manuscript addresses a very important issue with mapping GWAS risk loci to nearest genes, without considering the 3-dimensional organisation of the genome. Overall, the HiChIP data provided and performed analysis are convincing and have a potential to be useful to the field.
Major points:
1). The authors performed HiChIP in 3 endometrial cancer cell lines and a normal cell line and they have shown that putative target genes they identified are differentially expressed in tumours as compared to normal and associated with cancer-related pathways. Therefore, I would expect HiChIP loops connecting these target genes to be differential between normal and cancer cells? The authors should show if studied GWAS loci are associated with loss or gain of enhancer-promoter interactions, resulting in altered expression of connected genes.
2). Line 119: “…36 HiChIP target genes were found to be differentially expressed”. This result should be confirmed e.g. using Chi-square test to show if it's above what is expected by chance.
3). The manuscript would benefit from more figures showing representative examples of the identified associations i.e. HiChIP loops, CVs and gene expression changes.
4). Have the authors already submitted the raw data to GEO?
Author Response
Major points:
1). The authors performed HiChIP in 3 endometrial cancer cell lines and a normal cell line and they have shown that putative target genes they identified are differentially expressed in tumours as compared to normal and associated with cancer-related pathways. Therefore, I would expect HiChIP loops connecting these target genes to be differential between normal and cancer cells? The authors should show if studied GWAS loci are associated with loss or gain of enhancer-promoter interactions, resulting in altered expression of connected genes.
Of the 36 GWAS target genes that were differentially expressed, 4 were found by a gain of enhancer-promoter interactions in the normal cell line, 8 from enhancer-promoter interactions present in the normal cell line and least one cancer cell line, and 24 from enhancer-promoter interactions gained in a cancer cell line (Supplementary Table 2). Using a Fisher’s exact test, we examined the proportions of differentially expressed genes from those identified from a gain of an interaction in the normal or a cancer cell line compared with those present in both the normal and a cancer cell line(s). From this analysis, we observed no enrichment of differentially expressed genes among the genes that are differentially targeted in normal or cancer cell lines (OR=0.92, 95%CI 0.35-2.51, p=1). However, it is important to note that these target genes are likely subject to other differential interactions that are not associated with endometrial cancer risk CVs. Thus, although it is reasonable to hypothesize that loss or gain of enhancer-promoter interactions in endometrial cell lines may relate to differential expression in tumor, our data, which focuses on enhancer-promoter pairs that intersect with endometrial cancer risk CVs, suggests in this context that this is not the case. However, this question is an area of interest and we plan to assess the relationship between differential gene expression and H3K27Ac chromatin looping (not restricted to that associated with endometrial cancer risk CVs) in future studies.
2). Line 119: “…36 HiChIP target genes were found to be differentially expressed”. This result should be confirmed e.g. using Chi-square test to show if it's above what is expected by chance.
To demonstrate statistically significant enrichment, the data had been analysed by a Fisher’s exact test (as stated in lines 341-3) and the results were presented in the following sentence i.e. OR=2.39, 95% CI 1.59-3.59, p=6.08×10-05 (line 130). We have edited this sentence for clarity.
3). The manuscript would benefit from more figures showing representative examples of the identified associations i.e. HiChIP loops, CVs and gene expression changes.
We have added Supplementary Figure S1 to provide an example of a locus at which CVs are located in a looping anchor at a promoter region (i.e. shared promoter of WT1 and WT1-AS), but for which there is no looping from distal CVs. We have also provided Supplementary Figure S2, demonstrating promoter-associated chromatin looping from CVs at our most complex loci (i.e. 17q21.32 which contains 38 HiChIP target genes). Lastly, we have generated a plot of the most significantly differentially regulated genes (Supplementary Figure S3).
4). Have the authors already submitted the raw data to GEO?
Submission to GEO is currently underway. The manuscript will be updated with an accession number once provided by GEO.
Round 2
Reviewer 1 Report
In this revised version the authors have addressed the most relevant concerns. I think the paper is improved.